# Chronic Sildenafil Treatment Improves Vasomotor Function in a Mouse Model of Accelerated Aging

**DOI:** 10.3390/ijms21134667

**Published:** 2020-06-30

**Authors:** Keivan Golshiri, Ehsan Ataei Ataabadi, Renata Brandt, Ingrid van der Pluijm, René de Vries, A. H. Jan Danser, Anton Roks

**Affiliations:** 1Department of Internal Medicine, Erasmus Medical Center, 3015 GD Rotterdam, The Netherlands; k.golshiri@erasmusmc.nl (K.G.); e.ataeiataabadi@erasmusmc.nl (E.A.A.); r.devries@erasmusmc.nl (R.d.V.); a.danser@erasmusmc.nl (A.H.J.D.); 2Department of Molecular Genetics, Erasmus Medical Center, 3015 GD Rotterdam, The Netherlands; r.brandt@erasmusmc.nl (R.B.); i.vanderpluijm@erasmusmc.nl (I.v.d.P.); 3Department of Vascular Surgery, Erasmus Medical Center, 3015 GD Rotterdam, The Netherlands

**Keywords:** phosphodiesterase, sildenafil, aging, cGMP, nitric oxide, vascular dysfunction, hypertension, guanylate cyclase

## Abstract

Aging leads to a loss of vasomotor control. Both vasodilation and vasoconstriction are affected. Decreased nitric oxide–cGMP-mediated relaxation is a hallmark of aging. It contributes to vascular disease, notably hypertension, infarction, and dementia. Decreased vasodilation can be caused by aging independently from cardiovascular risk factors. This process that can be mimicked in mice in an accelerated way by activation of the DNA damage response. Genetic deletion of the DNA repair enzyme ERCC1 endonuclease in mice, as in the case of *Ercc1^Δ/-^* mice, can be used as a tool to accelerate aging. *Ercc1^Δ/-^* mice develop age-dependent vasomotor dysfunction from two months after birth. In the present study we tested if chronic treatment with sildenafil, a phosphodiesterase 5 inhibitor that augments NO–cGMP signaling, can reduce the development of vasomotor dysfunction in *Ercc1^Δ/-^* mice. *Ercc1^Δ/-^* mice and wild-type littermates were treated with 10 mg/kg/d of sildenafil from the age of 6 to the age of 14 weeks. Blood pressure and in vivo and ex vivo vasomotor responses were measured at the end of the treatment period. *Ercc1^Δ/-^* mice developed decreased reactive hyperemia, and diminished NO–cGMP-dependent acetylcholine responses. The diminished acetylcholine response involved both endothelial and vascular smooth muscle cell signaling. Chronic sildenafil exclusively improved NO–cGMP signaling in VSMC, and had no effect on endothelium-derived hyperpolarization. Sildenafil also improved KCl hypocontractility in *Ercc1^Δ/-^* mice. All effects were blood pressure-independent. The findings might be of clinical importance for prevention of morbidities related to vascular aging as well as for progeria patients with a high risk of cardiovascular disease.

## 1. Introduction

Cardiovascular disease (CVD) is a major cause of death globally, and related to aging in populations of developed countries. Age-related alterations in vascular function and morphology are leading causes of CVD [1,2,3]. It is believed that in its initial phase of development vascular aging is marked by decreased vasodilation due to impaired nitric oxide (NO)–cGMP signaling. This impairment partly depends on the balance between cGMP production and degradation. Soluble guanylyl cyclase (sGC) activation by NO, which is mainly formed by endothelial NO synthase (eNOS), plays a major role in cGMP production. Phosphodiesterases (PDEs) are responsible for cGMP degradation. Their increase impairs vasodilation and increases stiffness, blood pressure, hypertrophy, and inflammation [4,5]. During aging the metabolic balance turns toward cGMP degradation, and it has been shown that acute inhibition of PDE improves vasodilation in aged blood vessels [6,7,8].

PDEs form a family of enzymes that degrade the substrates cAMP, cGMP, or both. There are 11 PDE subtype families (1–11), and each member displays tissue and substrate specificity [9]. PDE5 is expressed in vascular smooth muscle cells (VSMC), where it plays a key role in regulation of cGMP, for which it is substrate-specific. Although it is well known that NO–cGMP signaling is decreased in aging, and that PDE5 inhibition might be used to improve this signaling pathway, this strategy is not being exploited to decelerate vascular aging. PDE5 inhibition currently addresses pulmonary arterial hypertension and erectile dysfunction. Although the latter is an expression of vascular aging, it is unclear if improvement of NO–cGMP signaling can counteract the aging process in the vasculature [10].

An important issue during exploration of this question was the duality in the etiology of vascular aging. Vascular aging features develop on the basis of two major causes. The first is the impact of physicochemical stimuli, including hypertension, inflammation, and hyperglycemia, that slowly worsen vascular function as they induce progressive, irreversible remodeling. Risk factor-reducing treatments, such as blood pressure, lipid, and glycemia lowering, have been reported to improve both NO–cGMP signaling and aging-associated vascular remodeling [4,11]. Conversely, inhibition of NO–cGMP signaling mimics such remodeling processes [12]. This gives the impression that NO–cGMP signaling regulates vascular aging and that risk factor-directed therapies are efficient. However, the vasculature is also affected by a second mechanism: An intrinsic aging process, which occurs independently from these stimuli, henceforth called ‘intrinsic vascular aging’ [4]. Whereas risk factor-dependent vascular aging has been the subject of intensive exploration for many decades, intrinsic vascular aging is hardly understood, and a confirmed treatment rationale is lacking.

During the last two decades it has become increasingly clear that the aging process that occurs intrinsically is a response to accumulating DNA damage [13,14]. Mice that are genetically modified to impair DNA repair age rapidly and develop main features of vascular aging in a much accelerated fashion in comparison to wildtype mice [6,7,15,16]. This includes the development of decreased NO–cGMP signaling, partly due to oxidative stress, increased blood pressure, and increased vascular stiffness. Previously we showed that *Ercc1^Δ/-^* mice, in which the function of the endonuclease ERCC1 was genetically disrupted to accelerate aging, develop all these features and can be used as a model to test interventions [6,7,17,18]. Restriction of food intake by 30%, a well-known anti-aging intervention, strongly increased longevity and health span in *Ercc1^Δ/-^* mice and preserved NO–cGMP signaling aging, something that cannot be achieved with blood pressure lowering and renin-angiotensin system blockade, a well-known risk-reducing treatment in hypertensive and diabetic patients [19,20]. We demonstrated that PDE5 might be involved in reduced vasodilation in *Ercc1^Δ/-^* mice [6]. Since deceleration of vascular aging with dietary restriction is associated with improved cGMP signaling, we asked if this increase could be responsible for the anti-aging effect. In this case, chronic PDE5 inhibitor treatment should decelerate intrinsic vascular aging. To test this hypothesis, we treated *Ercc1^Δ/-^* mice with sildenafil from very early age until the development of a progressed aging phenotype, and tested vasodilation capacity as a variable of vascular aging. In addition, we studied the fate of signaling through NO and endothelium-derived hyperpolarization (EDH). As a secondary objective we also studied contractile responses to KCl, which tend to be decreased in aging mice and humans [6,20,21,22].

## 2. Results

### 2.1. Unchanged Systolic and Diastolic Blood Pressure in Accelerated-Aging Model

Blood pressure measurements confirmed that at the age of 14 weeks blood pressure in *Ercc1^Δ/-^* and WT was not significantly different (Figure 1). Also, 8 weeks of treatment with sildenafil selectively reduced diastolic blood pressure in WT mice (Figure 1).

### 2.2. Reduced Cutaneous Vasodilation in Ercc1^Δ/−^ Mice

Reduced vasodilation in limbs is a well-known feature of vascular aging, which is at least partly dependent on eNOS signaling in the cutaneous vasculature where it can be measured [23,24]. To this purpose, we measured reactive hyperemia using a laser Doppler technique, using maximal response (Emax) and area under the curve over a 10-minute period after occlusion (AUC) as read-outs [6,17]. Emax showed a trend towards lowering in *Ercc1^Δ/-^* and AUC was significantly lower (*p* < 0.05) compared to WT (Figure 2). Sildenafil partly reversed the difference in AUC between *Ercc1^Δ/-^* and WT, and there was no significant difference between sildenafil-treated WT and *Ercc1^Δ/-^* (Figure 2a). A trend towards improvement in AUC by sildenafil vs. vehicle was observed within the *Ercc1^Δ/-^* group, although this did not reach statistical significance (Figure 2a).

### 2.3. Diminished Endothelium-Dependent and -Independent Response in Ercc1^Δ/−^ Mice

To explore the mechanism of the decreased vasodilator response, ex vivo wire myography experiments were performed in aortic rings with both the endothelium-dependent dilator ACh and the endothelium-independent dilator SNP. The ACh response was diminished in *Ercc1^Δ/-^* mice in comparison with their WT littermates (Figure 3a). There was also a significantly decreased vasorelaxation in *Ercc1^Δ/-^* mice in response to 0.1 mmol/L SNP (Figure 3c). The diminished relaxation response to ACh remained significant after correcting it for the difference in SNP response (Figure 3b). Taken together, these data indicated that there is both endothelial and smooth muscle dysfunction in the accelerated-aging mice.

To explore the signaling pathway underlying the decreased endothelium-dependent vasodilations, we studied the contribution of NO, EDH, and prostanoids to the ACh relaxation in aortic rings, using the NO inhibitor L-NAME, the EDH inhibitor cocktail TRAM34 and apamin, and the cyclo-oxygenase inhibitor indomethacin. However, since indomethacin exerted no blocking effects in WT mice, either alone or on top of TRAM34 and apamin (data not shown), and given that the size of *Ercc1^Δ/-^* mouse aorta was too small to study multiple conditions in parallel, we decided to focus on NO and EDH only in the sildenafil-treated mice. In both vehicle-treated *Ercc1^Δ/-^* and WT mice, L-NAME significantly decreased the ACh response (Figure 3d,e). Yet, the effect of L-NAME was less pronounced in *Ercc1^Δ/-^* than in WT. TRAM34/apamin had no effect in *Ercc1^Δ/-^* mice (Figure 3e) and non-significantly reduced the ACh response in WT mice on top of L-NAME. The residual response in the presence of both L-NAME + TRAM34/apamin was not different between *Ercc1^Δ/-^* and WT. Altogether, the results indicated that specifically the NO-mediated relaxations were lowered in *Ercc1^Δ/-^* vs. WT.

### 2.4. The Effects of Chronic Treatment with Sildenafil on Endothelium-Dependent and -Independent Response

Having established that NO-mediated vasodilations were decreased in *Ercc1^Δ/-^*, a typical vascular aging feature that was present in this model and in aging animals in general, the effect of chronic sildenafil was evaluated. Chronic treatment with sildenafil in drinking water did not improve the ACh response in WT animals. Yet, it did improve the vasodilatory response to ACh in *Ercc1*^Δ/-^ animals, albeit without restoring this to the level of WT (*p* < 0.0001, *Ercc1^Δ/-^* with sildenafil vs. WT with sildenafil).

To explore if sildenafil improved ACh-induced vasodilation through its effect on NO signaling in VSMC, the response to exogenous NO, provided by 0.1 mmol/L SNP, was evaluated. SNP responses were increased in *Ercc1^Δ/-^* by chronic sildenafil, indicating that the endothelium-independent responsiveness of VSMC to NO–cGMP was improved (Figure 4c). To further test if this also underlies the endothelium-dependent ACh response, the ACh response in *Ercc1^Δ/-^* mice was corrected for the response to 0.1 mmol/L SNP (Figure 4d). This approach eliminated the statistical significance of the effect of chronic sildenafil in *Ercc1^Δ/-^* mice, indicating that improved NO–cGMP signaling in VSMC mediates this effect. In agreement with this observation, L-NAME, but not TRAM34/apamin, reduced the ACh response in sildenafil-treated mice (Figure 5a,b). This simultaneously implies that sildenafil did not affect the EDH pathway.

### 2.5. Effect of Sildenafil on Vasoconstriction Mediated by KCl 100

Disturbed vasoconstriction is another feature of age-related vasomotor changes [6,20,21,22]. KCl was chosen to study alterations in contractility because it is free of receptor-mediated signaling. Acting directly through the opening of voltage-gated calcium channels, it is expected to involve less confounding factors on Ca^2+-^ mediated actin-myosin constriction. As expected, contractions to 100 mmol/L KCl were decreased in *Ercc1^Δ/-^* vs. WT mice (Figure 6). Sildenafil improved constrictions only in *Ercc1^Δ/-^* mice, although they remained smaller than those of WT (Figure 6).

## 3. Discussion

Cardiovascular disease (CVD) is the leading cause of death in people 65 years of age and older. Vascular aging plays an important role herein. This aging partly takes place independently of the presence of risk factors, due to the aging process that intrinsically occurs in any organism. To examine intrinsic vascular aging, we used a well-established accelerated-aging mice model, the *Ercc1^Δ/-^* mouse, which has been used previously to study the impact of aging on organ function and to develop potential remedies [25]. In the present study we explored the cardiovascular features and the contribution of NO, EDH, and residual endothelial signaling pathways in *Ercc1^Δ/-^* mice. In addition, we tested the hypothesis that chronic increase of NO–cGMP-signaling pathway by sildenafil can improve vasomotor function. We found that sildenafil improved vasodilator function measured in vivo and in vitro in *Ercc1^Δ/-^* mice, and that this effect takes place due to improved NO–cGMP signaling at the level of the VSMC. Sildenafil also improved vasoconstriction to KCl.

This is the first report to show for that PDE5 inhibition has a beneficial effect on vasomotor function in vivo in a mouse model of accelerated vascular aging. The effect of the chronic sildenafil treatment on SNP responses, which were fully mediated by cGMP in our mice [7], was similar in size to the acute effect of 100 nmol/L PDE5 found previously in an ex vivo organ bath experiment [7]. This demonstrated that in the present in vivo study sildenafil was administered in a dosage sufficiently high to reach the vascular wall. Also, our results warrant that chronic administration does not lead to disappearance of the sildenafil effect, a question that was raised shortly after the introduction of this medicine in the clinic [26]. Moreover, the effect extends to vasoconstriction, which is a new, unexpected finding. In addition, the effect on endothelium-dependent, EDH-mediated acetylcholine responses was examined. The results suggest that chronic sildenafil did not change EDH, which was not tested before in a model of aging.

We did not see any difference in blood pressure in the 14-week-old *Ercc1^Δ/-^* mice. This was also the case in 12-week-old *Ercc1^Δ/-^* mice [20], but not in 16-week-old *Ercc1^Δ/-^* mice, in which blood pressure was modestly increased [6]. The etiology of the increased blood pressure in *Ercc1^Δ/-^* mice is still unknown, and could arise both from the vascular aging phenotype and renal aberrations. In this article we confirmed that vasomotor dysfunction precedes blood pressure increase in *Ercc1^Δ/-^* mice, and might, therefore, contribute to its etiology. Although a strong renal phenotype is found in *Ercc1^Δ/-^* mice [27,28], the relation to blood pressure is not clear yet [17]. Nevertheless, it is very unlikely that sildenafil exerted effects through blood pressure in the present study; sildenafil did not change blood pressure in *Ercc1^Δ/-^* mice, but only in WT, where it did not change vasomotor function. Thus, sildenafil appears to act directly on the vasculature and not via blood pressure changes, which also dismisses a role of blood pressure regulation by the kidney in our present study. The modest effect of sildenafil on blood pressure, observed in WT, and absence thereof in *Ercc1^Δ/-^* resembles previous studies that showed a negligible blood pressure lowering effect of sildenafil under normal conditions as compared to a higher effectiveness during treatment with nitrites [29], which increases NO. Apparently, to be effective on blood pressure, sildenafil requires sufficient NO levels. The absence of the depressor effect in *Ercc1^Δ/-^* might point to the importance of reduced NO signaling in these accelerated-aging mice for blood pressure regulation. Together, the present data and those of our previous studies suggest that the increased blood pressure previously observed in older *Ercc1^Δ/-^* mice had a vascular cause. The fact that NO signaling must be strongly reduced before increased blood pressure can be observed can be explained by compensatory mechanisms, such as the adaptation of cardiac output, as we have previously shown [17]. Autonomic adaptation mechanisms might also have affected results measured with the laser Doppler technique; the effect of chronic sildenafil on reactive hyperemia was modest. On the other hand, the effect size of chronic sildenafil on ACh and SNP-mediated vasodilations was comparable to that on reactive hyperemia [7], which pleads for a major contribution of improved NO–cGMP signaling.

We recently demonstrated the acute capacity of sildenafil to augment NO–cGMP signaling ex vivo in aortic rings of *Ercc1^Δ/-^* mice, using the endothelium-independent NO donor SNP [7]. In the present study we reported for the first time the chronic effect of in vivo sildenafil treatment on aging-associated vasomotor dysfunction and the effect on both endothelium-dependent and -independent relaxations in parallel. Sildenafil did not restore the ACh response to normal, indicating that for this purpose increasing the cGMP half-life was not enough. Of note, also acute PDE inhibition did not fully restore SNP responses in *Ercc1^Δ/-^* mice [7]. A possible explanation for the incomplete reversal could be oxidation of the sGC heme group which would lead to decreased cGMP production, although no substantial difference was found in aortic cGMP production between *Ercc1^Δ/-^* and WT mice during PDE inhibition in a previous study [7]. The partial effect contrasts with food restriction, which fully normalized the ACh response by upregulating endothelial prostaglandin release, yet without altering the contribution of the NO pathway [19,20]. Clearly, future studies should now evaluate whether the combination of food restriction and sildenafil acts synergistically when compared to each treatment alone. Ideally, aging treatment restores both the endothelial and VSMC dysfunction.

An indication that sildenafil treatment may also act beyond NO–cGMP-dependent relaxation is its effect on hypocontractility. Hypocontraction, in addition to diminished relaxation, is a hallmark of vascular aging. Reduction of DNA repair capacity by targeted mutation of Lmna in VSMC, the gene that is responsible for accelerated-aging in Hutchinson–Gilford progeria, causes hypocontraction in mice. Interestingly, this aging feature can be counteracted by increased dietary nitrites [22]. Reduced vascular stiffness was proposed as a mechanism. However, it cannot be excluded that other processes, such as the contractile to synthetic phenotype switch of VSMC, another aging feature that might reduce constriction, are involved. During a previous study, of which the results were published [6], preconstrictions to KCl or U46619 were acutely decreased by sildenafil to the extent that NO inhibitor had to be added in the organ bath to warrant a sufficiently high constriction. This sharply contrasts with the increased KCl constriction found in aortic segments of sildenafil-treated *Ercc1^Δ/-^* mice in the present study, which was performed in the absence of L-NAME. Although an in-depth evaluation of vasoconstrictions, employing also other agonists, should shed further light on the mechanisms behind this observation, our present study provides compelling additional evidence that augmentation of NO signaling, which can be accomplished by increasing its availability (eg. through increased dietary nitrite) or decreasing breakdown of cGMP (through sildenafil, as in the present study) is effective when safeguarding proper contractile function during aging.

In summary, our work demonstrates the impact of sildenafil on intrinsic vascular aging in a mouse model of accelerated aging. It mimics the effect of food restriction on ACh-induced relaxation, albeit via a different mechanism, thus behaving as a treatment that might decelerate vascular aging, possibly in combination with food restriction. It also reverses hypocontractility. Since the pathways that connect the DNA damage response to vascular aging are not yet fully understood, sildenafil might be a novel tool to further increase our understanding of this phenomenon. Moreover, being a clinically applied drug, chronic sildenafil treatment is a potential pharmacologic strategy to reduce the risk of CVDs during aging or in progeria. The safety profile of sildenafil in older men that regularly use this drug for erectile dysfunction and in patients with pulmonary hypertension has been well described for chronic use [26,30]. Sildenafil is well tolerated, and adverse events were mostly transient and mild to moderate in severity. Caution has to be taken in patients taking antihypertensive medicines or nitrates because of the precipitation of hypotension. In general, though, chronic sildenafil treatment against vascular aging is a plausible possibility.

## 4. Materials and Methods

### 4.1. Animals

*Ercc1^Δ/-^* and *Ercc1*^+/+^ F1 mice with a hybrid C57BL6J: FVB background were generated by cross-breeding of parents with a pure C57BL6J and FVB background, as described before [31]. The hybrid background of the experiment mice prevented strain-specific phenotypes. Breedings were performed at the Erasmus MC animal facility. Mice were housed in individually ventilated cages in a controlled environment (20–22 °C, 12h light:12 h dark cycle) with access to normal chow and water ad libitum. The animals were weighed and visually inspected every day to monitor their well-being. All animal studies were performed in accordance with the Principles of Laboratory Animal Care and with the guidelines approved by the independent Dutch Animal Ethical Committee.

### 4.2. Study Design

In all, 24 *Ercc1^Δ/-^* mice at the age of 6 weeks old and 28 of their WT littermates (*Ercc1^+/+^*) at the same age were divided into four groups: One group of *Ercc1^Δ/-^* mice and one of the WT were given normal drinking water; another group of *Ercc1^Δ/-^* and WT mice received the PDE5 inhibitor (sildenafil 10 mg/kg/day) [32] via drinking water for 8 weeks. Both male and female mice were used: 28 mice in vehicle group (14 males and 14 females) and 24 mice in Sildenafil treated group (10 males and 14 females). We did not observe differences in response to the agonists used in the present experiments in untreated male and female mice of any of our previous studies [6,19,20]. Blood pressure and superficial blood flow were measured one week prior to sacrifice. At the age 14 weeks the mice were sacrificed and organ bath experiments were done.

### 4.3. Blood Pressure and Vasodilator Function (In Vivo)

A laser Doppler perfusion imaging system (Perimed, PeriScan PIM 3 System) was used to assess in vivo hind leg vasodilator function, as described before [6,7]. In short, mice were anesthetized by 2.8% isoflurane/O_2_ ventilation (Penlon, Sigma Delta vaporizer), while keeping the body temperature at 37.0 °C. The hind leg, while kept in a fixed position, was occluded for two minutes with a tourniquet. Upon release of the tourniquet, reactive hyperemia was measured for 10 min. Results were expressed as the maximum response or the area under the response curve (only the area above the baseline was considered).

The in vivo blood pressure was measured in conscious animals by an experienced technician, using the tail-cuff technique (CODA High-Throughput device from Kent Scientific) after five daily sessions: Four trainings sessions and a subsequent measurement session to record 40 measurement cycles. Invalid cycles were excluded by the CODA software, and the average of the valid cycles was used for comparisons [6,7].

### 4.4. Sacrifice and Wire Myography

The thoracic aorta was isolated and cleaned in cold oxygenated (with 95% O_2_ and 5% CO_2_) Krebs–Henseleit buffer solution (in mmol/L: NaCl 118, KCl 4.7, CaCl_2_ 2.5, MgSO_4_ 1.2, KH_2_PO_4_ 1.2, NaHCO_3_ 25, and glucose 8.3; pH 7.4) for ex vivo wire myography experiments. The 2-mm segments of thoracic aorta were mounted in 6-mL chambers of wire myography device (Danish Myograph Technology, Aarhus, Denmark). After a normalizing procedure [33], the maximum contractile responses were determined using 100 mmol/L KCl. After four times washing steps with 5 min interval for each step, 30 nmol/L U46619 was applied to preconstrict the vessel segments and evaluate the relaxation concentration-response curves (CRCs) to acetylcholine (ACh) and sodium nitroprusside (SNP) (respectively). L-NAME 100 μmol/L, TRAM34 10 μmol/L, and apamin 100 nmol/L were given 10 minutes before U46619, to investigate the involvement of nitric oxide (NO) and endothelium-dependent hyperpolarization (EDH) pathway in the relaxation responses. Indomethacin 10 μmol/L was given to determine the contribution of prostanoids to vasomotor responses.

### 4.5. Data Analysis

Relaxation to ACh and SNP was expressed relative to the contraction produced by U46619, which was set at 100% in each individual aortic ring. Data are shown as the percentage of relaxation, expressed as the mean ± S.E.M. The number of each individual experiment is shown for each of the rings. Statistical analysis was conducted using IBM SPSS statistics (IBM Corporation, version 25) and (GraphPad Prism, version 8.0.1; GraphPad Software Inc., San Diego, CA). Data were analyzed using Student’s paired t test and a general linear model repeated measurements. P values less than 0.05 were considered significant.

## Figures and Tables

**Figure 1 ijms-21-04667-f001:**
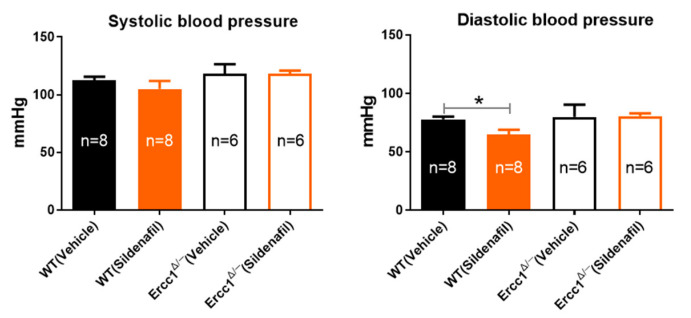
Systolic and diastolic blood pressure as measured with tail cuff in conscious *Ercc1^Δ/−^* mice versus their WT littermates. The number of mice per group is shown in the bars. Statistical differences were analyzed by two-tailed t-test between vehicle- and sildenafil-treated mice (*, *p* < 0.05).

**Figure 2 ijms-21-04667-f002:**
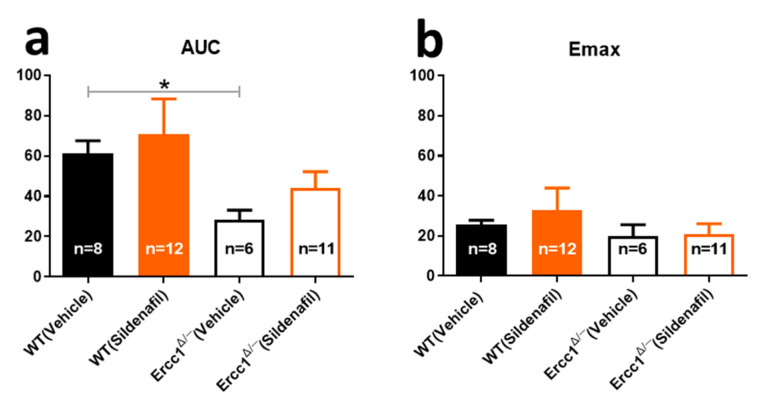
Cutaneous reactive hyperemia in *Ercc1^Δ/−^* mice versus their WT littermates measured with laser Doppler, expressed as: (**a**) AUC and (**b**) maximum response (Emax). The number of mice per group is shown in the bar. Statistical differences were analyzed by one-way ANOVA corrected for multiple testing (*, *p* < 0.05).

**Figure 3 ijms-21-04667-f003:**
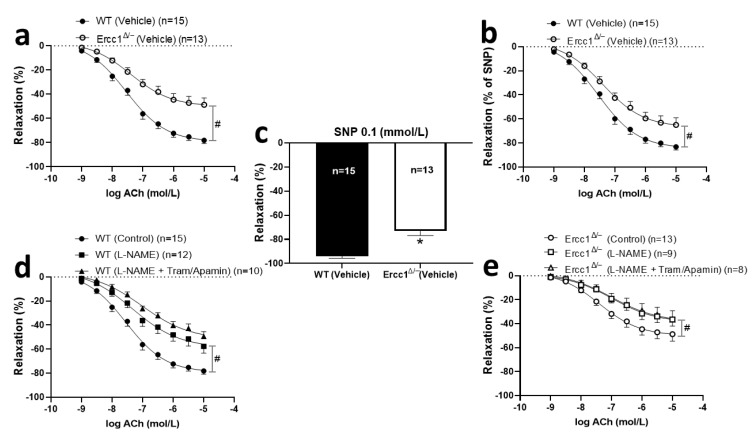
Endothelium-dependent (**a**) and -independent (**c**) responses and endothelium-dependent response corrected by SNP (**b**) in isolated aortic rings from *Ercc1^Δ/−^* mice (open circles) versus rings from WT littermate (filled circles), measured ex vivo in small wire organ baths. The impact of the NO synthesis inhibitor L-NAME (100 μmol/L) and EDH inhibitors TRAM34 (10 μmol/L) and apamin (100 nmol/L) (**d**,**e**) on the relaxations induced by ACh in aortic rings precontracted by U46619. Relaxations were calculated relative to the contraction produced by U46619 in each ring, which was set at 100%. Values are expressed as means ± SEM; n, number of mice; *, *p* < 0.05, t-test; #, *p* < 0.05; GLM for repeated measures.

**Figure 4 ijms-21-04667-f004:**
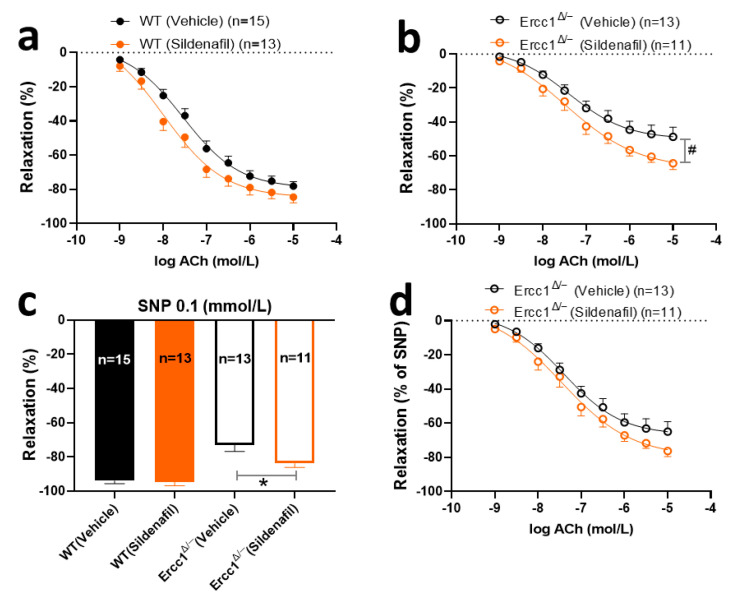
The effects of 8 weeks of treatment with sildenafil in drinking water on endothelium-dependent (**a**,**b**) and -independent responses (**c**) in *Ercc1^Δ/−^* mice (**b**,**c**) and WT littermates (**a**,**c**), measured in aortic rings in ex vivo organ bath experiments. Reponses to Ach in *Ercc1^Δ/−^* mice corrected for SNP (**d**). Data were calculated relative to the precontraction produced by U46619 in each ring. Values are expressed as means ± SEM; n, number of mice; *, *p* < 0.05, t-test; #, *p* < 0.05, GLM for repeated measures.

**Figure 5 ijms-21-04667-f005:**
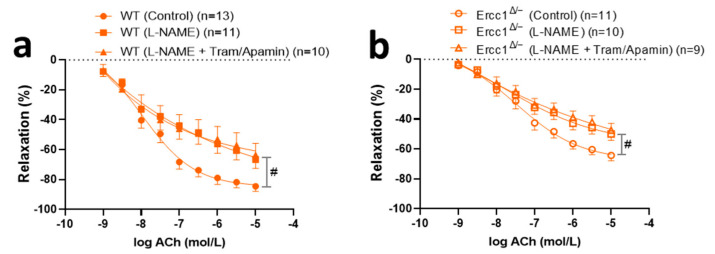
The impact of the NO synthesis inhibitor L-NAME (100 μmol/L) and EDH inhibitors TRAM34 (10 μmol/L) and apamin (100 nmol/L) on the relaxations induced by ACh in aortic rings precontracted by U46619 in *Ercc1^Δ/−^* mice (**b**) and WT littermates (**a**) treated with sildenafil. Data were calculated relative to the maximal changes from the contraction produced by U46619 in each ring, which was taken as 100%. Values are expressed as means ± SEM; n, number of mice; #, *p* < 0.05, GLM for repeated measures.

**Figure 6 ijms-21-04667-f006:**
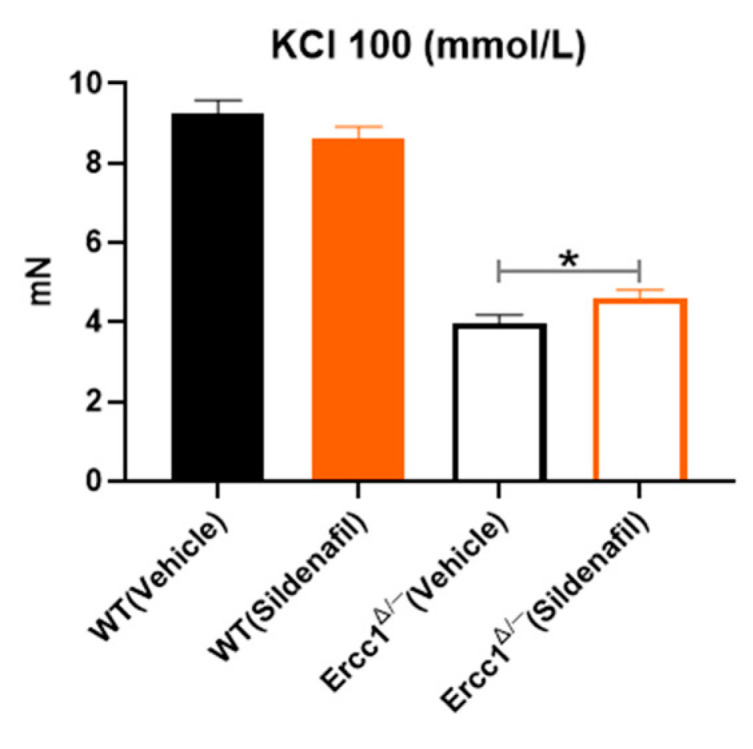
Maximum contraction response of aortic rings from *Ercc1^Δ/−^* mice versus their WT littermates to 100 mmol/L KCl. Statistical differences were analyzed by two-tailed t-test (*, *p* < 0.05).

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
