# Peer review of "Chronic Sildenafil Treatment Improves Vasomotor Function in a Mouse Model of Accelerated Aging"

_ijms, 2020, doi:10.3390/ijms21134667_

Round 1

Reviewer 1 Report

In this manuscript, Golshiri et al propose that inhibition of PDE5 by chronic administration of sildenafil improves vascular smooth muscle function in a transgenic model with vasomotor dysfunction induced by the deletion of ERCC1.

- Authors have already demonstrated that acute administration of sildenafil improves vasodilatation in aortic rings from Ercc1 mice. In their new set of data, they evaluate the efficacy of chronic administration of sildenafil in preventing vascular dysfunction in Ercc mice. The authors should emphasize the new mechanistic information with respect to their previous studies.

- Authors propose in the discussion section chronic administration of sildenafil as a pharmacological strategy to reduce vasomotor dysfunction during aging. However, this idea is not well supported by the present data:

1. Reactive hyperemia was used to evaluate the reduced vasodilatation associated with vascular ageing. Data shows decreased hyperemia in Ercc1 mice compared to WT. However, sildenafil shows limited effect in the intact animal. The authors should measure cGMP levels in these animals.

2.Even accepting that sildenafil prevents vasomotor dysfunction in Ercc1 transgenic mice, the data do not prove that the same effect would occur in aged mice. It has been proposed that oxidative stress increases in ageing. Oxidative stress is associated with eNOS uncoupling and oxidation of the prosthetic heme group of sGC. The efficacy of sildenafil in a condition of reduced cGMP production is unclear.

3.In addition, it has to be considered the described side effects of chronic administration of PDE inhibitors.

- Data should be analysed using two-way ANOVA to assess main effects, with mouse strain and sildenafil as the independent variables.

Reviewer 2 Report

The author Keivan Golshiri and the colleagues demonstrated that PDE5 inhibition attenuates vasomotor dysfunction in Ercc1/- mice. Although multiple observations and experiments were made, it is lack of convincing evidence to support the current conclusion, robust mechanism is needed as well.

Comment 1: Overall, there is a need to correct grammatical errors and improve English.  

Comment 2: Along with the vessel function studies, it is needed to measure levels of NO and cGMP by western blot or ELISA to solidate the findings.

Comment 3: For figure 6, it is not appropriate to conclude contraction is improved in sildenafil treated mice. More vasoactive agonist need to be applied including PE, U-46619 etc.

Comment 4: It is needed to describe the mice used in the study were male or female.

Round 2

Reviewer 1 Report

The authors have addressed my main concerns.

Reviewer 2 Report

The response to the queries of the reviewer was useful
and addition to the text was appropriate.